# Optimization of Callus and Cell Suspension Cultures of *Lycium schweinfurthii* for Improved Production of Phenolics, Flavonoids, and Antioxidant Activity

**Diaa Mamdouh** [1,2,*] and **Iryna Smetanska** [1,*]

1   Department of Plant Food Processing, Agricultural Faculty, University of Applied Sciences Weihenstephan-Triesdorf, Markgrafenstr 16, 91746 Weidenbach, Germany
2   Botany & Microbiology Department, Faculty of Science, Al-Azhar University, Nasr City, Cairo 11884, Egypt
*   Correspondence: diaa.mamdouh@azhar.edu.eg (D.M.); iryna.smetanska@hswt.de (I.S.)

**Abstract:** *Lycium schweinfurthii* is a traditional medicinal plant grown in the Mediterranean region. As it is used in folk medicine to treat stomach ulcers, it took more attention as a source of valuable secondary metabolites. The in vitro cultures of *L. schweinfurthii* could be a great tool to produce secondary metabolites at low costs. The presented study aimed to introduce and optimize a protocol for inducing callus and cell suspension cultures as well as estimating phenolic, flavonoid compounds, and antioxidant activity in the cultures of the studied species. Three plant growth regulators (PGRs) were supplemented to MS medium solely or in combination to induce callus from leaf explants. The combination between 2,4-dichlorophenoxy acetic acid (2,4-D) and 1-naphthyl acetic acid (NAA) induced callus in all explants regardless of the concentration. The highest fresh weight of callus (3.92 g) was obtained on MS medium fortified with 1 mg L$^{-1}$ of both 2,4-D and NAA (DN1) after 7 weeks of culture. DN1 was the best medium for callus multiplication regarding the increase in fresh weight and size of callus. Otherwise, the highest phenolics, flavonoids, and antioxidant activity against DPPH free radicals were of callus on MS fortified with 2 mg L$^{-1}$ NAA (N2). The cell suspension cultures were cultivated on a liquid N2 medium with different sucrose concentrations of 5–30 g L$^{-1}$ to observe the possible effects on cells' multiplication and secondary metabolite production. The highest fresh and viable biomass of 12.01 g was obtained on N2 containing 30 g L$^{-1}$ sucrose. On the other hand, the cell cultures on N2 medium of 5 and 30 g L$^{-1}$ sucrose produced phenolics and flavonoids, and revealed antioxidant activity against DPPH and ABTS$^+$ free radicals more than other sucrose concentrations. The presented protocol should be useful in the large-scale production of phenolic and flavonoid compounds from callus and cell cultures of *L. schweinfurthii*.

**Keywords:** ABTS; DPPH; dedifferentiation; in vitro cultures; *L. schweinfurthii*; medicinal plants; secondary metabolites

## 1. Introduction

The *Lycium* genus is a widespread Solanaceae family member comprising about 97 species. The most well-known *Lycium* species are *Lycium barbarum* and *L. chinense*. These are commonly used for their antioxidant activity and health properties [1]. The functional effects of a plant species depend on active constituents, namely the secondary metabolites. The higher content of active secondary metabolites, the more they are used in folk medicine, and the higher the effect on human health [2,3]. *L. schweinfurthii* is a Solanaceae member which is used in folk medicine and distributed in the Middle East region. It was reported that leaves and fruits of *L. schweinfurthii* are traditionally used to treat stomach ulcers [4]. A preliminary phytochemical screening in different *L. schweinfurthii* plant parts showed the presence of alkaloids, glycosides, sterols, saponins, resins, phenolics, and flavonoids. Furthermore, the total flavonoids and other secondary metabolites were concentrated in the leaves more than in the other plant parts [5,6]. Leaves of *L. schweinfurthii* were reported

to have rutin as a flavonoid glycoside, while stems have $\beta$-sitosterol and diosgenin as steroids [7,8]. Twenty-eight compounds were isolated from *L. schweinfurthii* during a study of its cytotoxicity effect. All of them showed minimal toxicity towards normal cells from the skin and colon, indicating their potential selectivity and safety as cytotoxic compounds. Four isolated compounds showed cytotoxic effects against skin cancer cells, while three of the isolated flavonoids showed potent cytotoxic activity against colon cancer cells [9]. Moreover, four glucosides that have a lowering effect in postprandial hyperglycemia in diabetic patients were isolated from *L. schweinfurthii* [10].

Plant cell and tissue culture are biotechnological techniques that play an essential role in the production of active compounds and in exploring alternative pharmaceuticals from plants [11]. Successful plant tissue culture depends on many factors, particularly plant growth regulators (PGRs), the age of tissue, the type of medium, and the type of used explant [12,13]. From different plant tissue culture types, the callus and cell suspension cultures are familiar techniques for the production of a wide range of phytochemicals. The induction of callus, i.e., the undifferentiated cells produced from plants, could be of immense importance when it is used to produce and accumulate pharmaceuticals, including therapeutic and antioxidant compounds [14,15]. Callus cultures usually take a brief time to multiplicate, need small spaces, are not expensive, do not need special equipment, and could be applied on different scales.

One way the plants face environmental challenges and maintain their survival is the production of secondary metabolites [16]. A major group of plant secondary metabolites is phenolic compounds. They can be used as food additives, nutraceuticals, and pharmaceuticals, as well as influencing flavor, color, and texture [17]. Flavonoids are a group of phenolic secondary metabolites which are found in most plant species. Flavonoids have a counter effect against unfavorable environmental conditions such as drought [18], high concentrations of aluminum in soil [19], and UV-irradiation [20]. Moreover, the production of flavonoids acts as a defense mechanism in plants against herbivores, bacteria, and fungi [21]. Flavonoids are familiar as antioxidants and potential metal chelators [22,23]. Flavonoids have several therapeutic and disease inhibition properties which put them among the most attractive and interesting natural groups of active compounds for human nutrition and healthcare [24]. Protection of numerous diseases could be achieved by including flavonoids in the human diet due to their strong action as antioxidants [25], anti-inflammatory [26], anticarcinogens [27], antivirals [28], and antibacterials [29]. Flavonoids have a direct cytoprotective effect on coronary and vascular systems, as well as the liver and pancreas [30,31].

The biotechnological methods, particularly plant cell cultures, are attractive alternative sources to the whole plants to produce specific secondary metabolites with significant amounts [32]. Several reports have been conducted about the in vitro production of flavonoids using different biotechnological methods, including callus and cell suspension cultures [33]. The production of phenolic and flavonoid compounds was reported in micropropagated plants of *L. schweinfurthii* [34]. Obtaining phenolic and flavonoid compounds from the plant callus cultures is a common tool in recent decades, such as in *Plantago ovata* Forsk. [35], *Rhodiola rosea* L. [36], and *Rosmarinus officinalis* L. [37]. On the other hand, the cell suspension culture is a mutual technique for enhancing phenolics and flavonoids production with elicitors such as in *Salvia nemorosa* [38], *Vitis vinifera* [39], *Thevetia peruviana* [40], and *Origanum vulgare* [41]. Furthermore, the callus is the real start for the large-scale production of secondary metabolites through cell suspension cultures. Gai et al. [42] established cell suspension cultures of *Cajanus cajan* (Linn.) Millsp. for effective production of pharmaceutically active phenolic compounds. Moreover, Açıkgöz [43] studied the role of sorbitol in the production of phenolic compounds in the cell suspension cultures of *Ocimum basilicum* L.

On the account of the importance of searching for natural alternatives to the nutritional supplements needed by humanity, we aimed in this study to establish and present the

protocols of callus and cell suspension cultures of *L. schweinfurthii* as alternative sources of phenolic and flavonoid compounds with antioxidative activity.

## 2. Materials and Methods

### 2.1. Plant Material and Culture Conditions

The fresh fruits of *L. schweinfurthii* were collected and air-dried for 5 days until complete drying. The fruits envelopes were removed, and the seeds were subjected to sterilization using 70% ethanol for 30 s then 30% of commercial bleach for 10 min. To remove the remaining bleach, seeds were washed 4 times with deionized distilled water. The sterilized seeds were cultured in 250 mL flasks containing 50 mL basal MS (Murashige and Skoog) medium (Duchefa, Haarlem, The Netherlands) [44] containing 30 g $L^{-1}$ sucrose and solidified with 7 g $L^{-1}$ Agar (Duchefa, Haarlem, The Netherlands). The cultures were incubated at room temperature under a 16 h photoperiod with light intensity of 33.73 $\mu mol^{-1}$ $m^{-2}$ $s^{-1}$ by cool fluorescent lamps.

### 2.2. Callus Induction and Multiplication

Leaves of eight week old aseptic seedlings were used as explants source for callus obtainment. Three plant growth regulators were expected to be effective in calli production from such plant leaves—$N^6$-Benzyladenine (S.D.Fine-Chem Ltd., Mumbai, India) as a cytokinin while 1-Naphthyl Acetic Acid (Mallinckrodt, Hazelwood, MO, USA) and 2,4-Dichlorophenoxy acetic acid (Blulux, Faridabad, India) were auxins. Six replicates of apparently equal-sized pieces of the plant leaves were cultured on full-strength MS medium fortified with different concentrations of 2,4-D, NAA, and BA in addition to MS medium free of PGRs and incubated with the same physical conditions (Table 1).

**Table 1.** Concentrations of PGRs supplemented to MS medium to induce callus from *L. schweinfurthii* leaves.

| PGR | Concentration (mg $L^{-1}$) | | | | | | | | | | | | | | | | | | |
|---|---|---|---|---|---|---|---|---|---|---|---|---|---|---|---|---|---|---|---|
| **2,4-D** | - | 0.5 | 1 | 2 | - | - | - | - | - | - | 0.5 | 1 | 2 | - | - | - | 0.5 | 1 | 2 |
| **NAA** | - | - | - | - | 0.5 | 1 | 2 | - | - | - | - | - | - | 0.5 | 1 | 2 | 0.5 | 1 | 2 |
| **BA** | - | - | - | - | - | - | - | 0.5 | 1 | 2 | 1 | 1 | 1 | 1 | 1 | 1 | - | - | - |

One gram of callus was sub-cultured on solidified MS medium supplemented with 2,4-D (1 or 2 mg $L^{-1}$), NAA (1 or 2 mg $L^{-1}$), or 2,4-D+NAA (0.5 + 0.5 or 1 + 1 mg $L^{-1}$) for multiplication. Changes in size and fresh weights of callus were obtained every 4 days for 32 days incubation period. The callus area was measured using ImageJ Software as a measure for size enlargement in 5 replicates of each treatment. The fresh weights of callus were measured in 4 replicates of each treatment. The initial inoculation weight, the texture, and the age of the callus were fixed in all treatments.

### 2.3. Cell Suspension Cultures

The calli produced was sub-cultured in liquid MS medium supplemented with 2 mg $L^{-1}$ NAA to produce cell cultures. Different sucrose concentrations of 5, 10, 15, 25, and 30 g $L^{-1}$ were used to compare the differences in the fresh weight of the produced cells, the cell viability, the changes in medium pH, and secondary metabolites production. The pH of the used media was adjusted to 5.7 during preparation. Fifty milliliters of each medium were placed in 100 mL flasks and inoculated with 1 g friable creamy callus. Four replicates for each treatment were used. The cultures were cultivated in an Incubator Hood TH 30 temperature-controlled shaker (Edmund Bühler GmbH, Bodelshausen, Germany) at 28 °C with a shaking rate of 120 rpm for 2 weeks. At the end of the culture, the cell cultures were vacuum filtered using a PC 3001 VARIO® vacuum pump (Vacuubrand, Wertheim, Germany) and weighed. The pH of the remaining media was measured using a PH 20 pH meter (VWR, Darmstadt, Germany). The biomass of cell cultures was tested immediately for viability and stored at −20 °C for further analysis.

### 2.4. Cell Viability Test

The cell viability test with 2,3,5-triphenyl tetrazolium chloride (TTC) for samples of cell suspension cultures was applied to six replicates of cell cultures with each sucrose concentration. The method was adapted from CABRI (Common Access to Biological Resources and Information) guidelines of reference number PC/1998/2/3.4 with some modifications [45]. A fifty mg of vacuum-filtered cell culture (fresh weight) was put into a 5 mL Eppendorf tube and 500 µL of TTC solution (1% of TTC in TRIS-HCl buffer 0.05 M, pH 7.5) was added to the cells. The suspension was gently centrifuged after 6 h of incubation in the dark at room temperature and the reaction mixture was removed. Three ml of ethanol was added to the cells and then incubated overnight at room temperature. The suspension mixture was centrifuged gently to get the cell-free supernatant. The absorbance of the supernatant was measured with an Analytic Jena Specord® 250 Plus UV-Vis spectrophotometer (Jena, Germany) at 500 nm. If the $A_{500}$ is less than 0.05, the cell cultures are non-viable while they have low viability if the $A_{500}$ is between 0.05 and 0.15. The cells are considered viable in case of $A_{500}$ higher than 0.15 after 2 weeks of culture.

### 2.5. Sample Preparation and Extraction

Fresh samples of 2.5 g of callus and cell cultures were weighed and extracted with 5 mL preheated 80% aqueous methanol for 45 min in a sonicated water bath at a temperature of 80 °C, frequency of 37 kHz, and power of 60% using Elmasonic P 30 H (Elma, Singen, Germany). The samples were centrifuged for 5 min at 4000 rpm using Centrifuge 5702 (Eppendorf, Hamburg, Germany), and the supernatant was collected and increased up to 5 mL to get a 500 mg mL$^{-1}$ extraction solution for each sample. The extraction was done to all samples resulting from different callus multiplication media as well as cell culture liquid media. The total phenolic and flavonoid contents, as well as the antioxidant activity, were then estimated in the extraction solutions.

### 2.6. Total Phenolic Assay

The total phenolic content of callus and cell cultures was determined by using the Folin-Ciocalteu assay, as described by Mamdouh et al. [34]. Gallic acid (Sigma-Aldrich, St. Louis, MO, USA) was used for generating a standard curve using standard solutions of 10, 20, 30, 40, 50, 100, 150, 200, and 300 µg ml$^{-1}$. In a 5 mL Eppendorf tube, an aliquot (200 µL) of extracts or gallic acid standard solutions was mixed with 1.8 mL of distilled deionized water. To the mixture, 200 µL of Folin-Ciocalteu's reagent (Merck, Schnelldorf, Germany) was added, well mixed, and incubated at room temperature for 5 min. At the sixth minute, 2 mL of 7% sodium carbonate (VWR Chemicals, Darmstadt, Germany) solution was added and mixed thoroughly. The mixture was diluted to 5 mL with deionized distilled water, shaken, and kept for 90 min in the dark at room temperature. The absorbance against the reagent blank was determined at 750 nm with Analytic Jena Specord® 250 Plus UV-Vis spectrophotometer. Total phenolic content was expressed as µg g$^{-1}$ in fresh samples in equivalence to gallic acid.

### 2.7. Total Flavonoid Assay

Total flavonoid content was measured by aluminum chloride assay as described by Mamdouh et al. [34]. Catechin (Sigma-Aldrich, Louis, MO, USA) was used for generating a standard curve using standard solutions of 20, 40, 60, 80, 100, and 120 µg mL$^{-1}$. In a 5 mL Eppendorf tube, an aliquot (500 µL) of extracts or catechin standard solutions was mixed with 2 mL deionized distilled water. To the mixture, 150 µL of 5% sodium nitrite (AppliChem, Darmstadt, Germany) was added. Then, 150 µL of 10% aluminum chloride (Carl-Roth, Karlsruhe, Germany) was added after 5 min. At the sixth min, 1 mL of 1 M sodium hydroxide was added, and the total volume was made up to 5 mL using deionized distilled water. The absorbance was measured against the reagent blank at 510 nm. Total flavonoid content was expressed as µg g$^{-1}$ in fresh samples in equivalence to catechin.

### 2.8. Antioxidant Capacity

The antioxidant activity of callus and cell suspension cultures was measured using the DPPH (diphenyl-1-picryl-hydrazyl) assay according to Olalere et al. [46] and the ABTS (2,2′-azino-bis(3-ethylbenzothiazoline-6-sulphonic acid)) assay according to Gabr et al. [47]. Because of the higher water content in callus relative to the filtered cell cultures, different diluted extracts were used to estimate the antioxidant activity. Extraction concentrations of 25 mg mL$^{-1}$ of callus and cell cultures were used to compare the effect of different PGRs on the antioxidant activity of callus extracts and different sucrose concentrations on the antioxidant activity of extracts.

### 2.9. DPPH Antioxidant Capacity

DPPH stock solution of 1 mM was prepared in methanol and stored at $-20\,^{\circ}$C. The working solution of 0.06 mM was prepared and absorbance of $0.8 \pm 0.02$ at 515 nm was obtained. DPPH solution contains stable free radicals which convert to stable molecules by accepting hydrogen radicals or electrons. An aliquot (0.5 mL) of diluted extracts and blank (methanol instead of extract) were mixed with 2.5 mL of DPPH working solution. The mixture was kept in the dark for 30 min at room temperature and the absorbance was measured at 515 nm. The DPPH radical-scavenging activity percentage was calculated according to the equation $((A_0 - A_1)/A_0) \times 100$, where $A_0$ is the absorbance obtained in the blank and $A_1$ is the absorbance measured in the extracts.

### 2.10. ABTS Antioxidant Capacity

ABTS solution of 7mM was reacted with 2.4 mM potassium persulphate solution with a ratio of 1:1 (*v/v*) and incubated in the dark at room temperature for 16 h to prepare the free radicals of ABTS$^+$. One milliliter of the prepared ABTS$^+$ solution was mixed with 60 mL methanol to prepare a working solution with an absorbance of $0.60 \pm 0.02$ at 734 nm. An aliquot of 40 μL of diluted extracts and blank (methanol used instead of extract) were mixed with 1.96 mL of the ABTS$^+$ working solution and incubated in the dark for 10 min at 37 $^{\circ}$C and the absorbance was measured at 734 nm. The ABTS$^+$ radical-scavenging activity percentage was calculated according to the equation $((A_0 - A_1)/A_0) \times 100$ where $A_0$ is the absorbance obtained in the blank and $A_1$ is the absorbance measured in the extract's samples.

### 2.11. Recording Data and Statistical Analysis

Callus induction percentage was recorded after five weeks of cultivation while fresh weight was recorded after seven weeks. During callus multiplication, the fresh weights and size were recorded in 4 days intervals for 32 days. The fresh weights of cell cultures and pH of the medium were estimated at the beginning and end of the 2 weeks culture period. The obtained data were subjected to statistical analysis of variance (ANOVA) using SigmaPlot ver. 12.5 (Systat Software Inc., San Jose, CA, USA) where the pairwise comparison was done using the Holm–Sidak method.

## 3. Results and Discussion

### 3.1. Callus Induction and Multiplication

Sterilized plants of *L. schweinfurthii* were obtained after 3 weeks of seeds sterilization and germination on MS medium containing 30 g L$^{-1}$ sucrose, solidified with 8 g L$^{-1}$ Agar and free of PGRs (Figure 1). To initiate callus induction from leaf explants, equal-sized cut leaves were inoculated in callus induction media. The callus growth was initiated within 7–12 days and then increased in mass gradually. MS medium is the most common basal medium which is used in addition to PGRs to initiate callus from different explants of most plant species [48,49]. The callus initiation period differs according to the plant species and culture media from days to a few weeks. The callus of the medicinal plant *Caesalpinia bonducella* F. was initiated after 7 days on MS medium fortified with 2,4-D [50], while after 10 days in cultures of *Lindernia madayiparense* [51] and *Onosma bulbotrichom* [52].

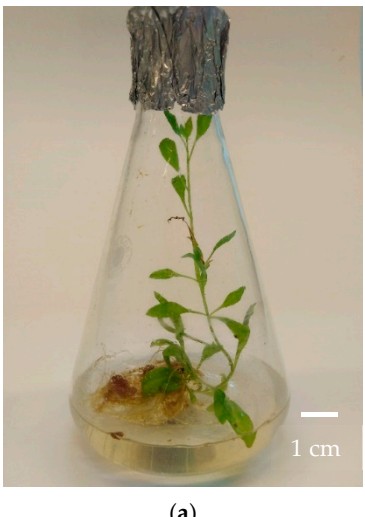
(**a**)

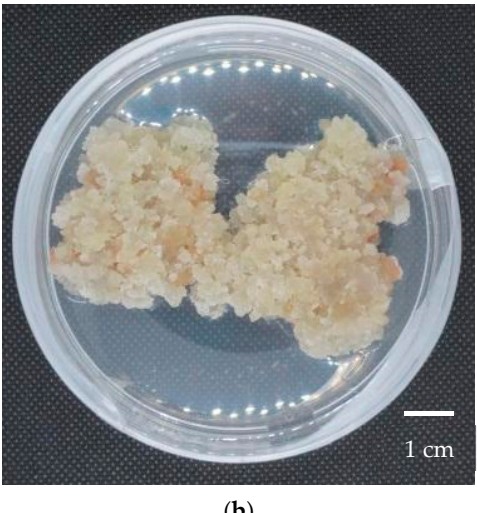
(**b**)

**Figure 1.** Callus induction of *L. schweinfurthii*. (**a**) Sterilized plants after 3 weeks of aseptic seeds germination; (**b**) Callus induced on MS medium fortified with 2 mg L$^{-1}$ NAA after 5 weeks of culturing.

The success rate of callus initiation percentage from leaf explants was associated to a great limit with the type and concentration of auxin used where the full callus induction was observed with auxins of 1 mg L$^{-1}$ and slightly lower with 2,4-D or slightly more with NAA. However, it was important to compare this productivity when BA was added as a cytokinin, where very few calluses formed when auxin and cytokinin were used together except in the case of a 2:1 ratio, respectively. Otherwise, the best callus induction from leaf explants was on using equal concentrations of both auxins used and without any BA addition (Figure 2). The relation between the high callus initiation percentage and type of auxin in *Triticum aestivum* L. [53], *Citrus grandis* [54], *Michelia champaca* [55], *Manihot esculentum* Cranz [56] has been recorded. It was found in the current study that the use of equal concentrations of the two auxins has a remarkable effect on the productivity of the callusm, although the use of NAA competes on the front but failed in this, except when its concentration was high. This may have been due to variations in the mechanism of action of both auxins to stimulate cell division. Auxins are necessary for the onset of callus growth [57] and their concentrations promote the total weight of callus produced [58].

Loose and creamy callus was observed in most cultures, while granular and friable callus was distinguished in combinations of NAA and 2,4-D and higher concentrations of NAA. The highest significant fresh weight of 3.92 g was recorded after 7 weeks of inoculating leaf discs on MS medium supplemented with an equal concentration of both auxins NAA and 2,4-D of 1 mg L$^{-1}$, while it was non-significant using 2 mg L$^{-1}$ NAA solely of 2.85 g fresh callus (Figure 1). On the other hand, it is noticed that in cultures containing BA solely or combined with auxin (2,4-D or NAA) with equal concentration, the leaves were enlarged and a little mass of callus formed. Furthermore, the enlarged leaves differentiated into new shoots directly, especially when using BA 0.5 or 1.0 mg L$^{-1}$ (Figure S1). The callus yield was associated with NAA concentration and increased significantly when used in conjugation with 2,4-D. The efficiency of callus induction is directly affected by the constituents of culture media, particularly the plant growth regulators. The fundamental PGRs in most callus cultures' media are the auxins, as they take a role in cell division and elongation. Otherwise, the combination of cytokinins with auxins in some cases could enhance the proliferation of callus significantly [58,59]. Moreover, the used explant type could produce significant differences in callus initiation. In a study for enhancement of total phenolic content of *Lycium barbarum* L. in vitro cultures, the hypocotyl explant exhibits more availability for callus initiation than leaf and root explants [60].

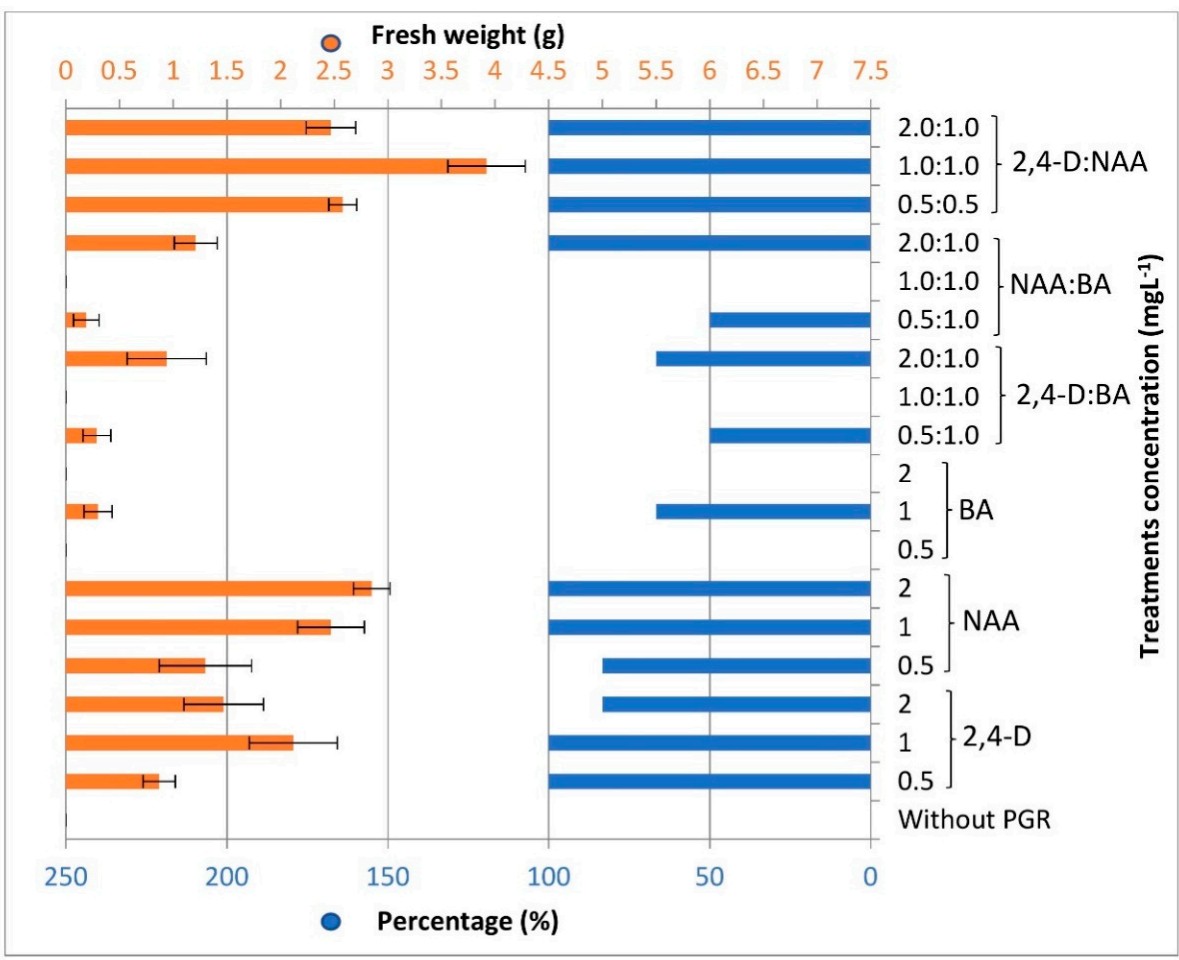

**Figure 2.** Callus induction percentage and fresh weights of callus induced from *L. schweinfurthii* leaves.

Six media were selected to study the callus multiplication of *L. schweinfurthii* according to the highest callus induction and fresh weights produced in the preliminary induction process. NAA or 2,4-D was supplemented solely with a concentration of 1 mg L$^{-1}$ (N1), 2 mg L$^{-1}$ (N2) of NAA, or 1 mg L$^{-1}$ (D1), and 2 mg L$^{-1}$ (D2) of 2,4-D. A combination of 2,4-D and NAA was supplemented to solidified MS medium with a concentration of 0.5 + 0.5 mg L$^{-1}$ (DN0.5) and 1 + 1 mg L$^{-1}$ (DN1). The callus fresh weights and area were measured every 4 days during the whole cultivation period of 32 days. A constant weight of 1 ± 0.05 g of friable creamy callus of the same age was used as the initial inoculum for the four replicates of each treatment. The obtained results showed that significant differences in callus fresh weights started to appear after 12 days of culturing, where the N1 medium was significantly higher than other media. The callus fresh weights on N1 were the highest in comparison with other treatments until 20 days of culture. The fresh weights of callus on N1 were 2.39, 2.88, and 3.05 g after 12, 16, and 20 days, respectively. Contrarily, the highest fresh weights of callus after 24 and 28 days of cultivation were obtained on DN1 medium of 4.24 and 4.85 g, respectively. The difference of DN1 was significant with N1, D2, and DN0.5 on day 24, while it is significant only with D2 after 28 days. The fresh weights of callus after 32 days ranged between 3.97 and 5.42 g without any significant differences between the different media. The highest weight was obtained on D1 and DN1 media at 5.42 and 5.20 g, respectively. The lowest weight of 3.97 g was observed on D2 medium. The recorded results showed that the weights of callus on N1 and D2 media were increasing slowly during the first 28 days. During the last four days, the weight of callus increased by more than one gram in both cultures. The fresh weights in N2, DN0.5, and DN1 media were boosted remarkably starting from day 20 and continued increasing until the end of

the incubation period (Table 2; Figure 3 and Figure S2). The PGRs have a direct effect on the callus multiplication rate, where auxins have a key role in the process [61]. The results of this study were in accordance with Solangi et al. [62] in the point of the effect of auxins especially 2,4-D on the callus induction of sugarcane (*Saccharum officinarum* L.). Contrary to our results, a combination of auxin and cytokinin (2,4-D and kinetin) was the best for initiating callus of *Crataegus pseudoheterophylla* [63] and *Barringtonia racemosa* L. [64], while a combination of 2,4-D and BA produced more callus fresh weight, sevenfold higher than other tested PGRs treatments on *Cnidium officinale* Makino root explants [65]. Furthermore, the combination of NAA and BA was the best for the callus induction of *Stevia rebaudiana* Bertoni [66].

**Table 2.** Fresh weights of callus multiplied on MS medium fortified with different concentrations of NAA and 2,4-D.

| Medium | Fresh Weight (g), Mean $\pm$ SE | | | | | | | | |
|---|---|---|---|---|---|---|---|---|---|
| | Day 0 | Day 4 | Day 8 | Day 12 | Day 16 | Day 20 | Day 24 | Day 28 | Day 32 |
| N1 | 1.00 $\pm$ 0.00 [a] | 1.41 $\pm$ 0.03 [a] | 1.40 $\pm$ 0.04 [a] | 2.39 $\pm$ 0.08 [a] | 2.88 $\pm$ 0.19 [a] | 3.05 $\pm$ 0.39 [a] | 3.29 $\pm$ 0.08 [bc] | 3.40 $\pm$ 0.28 [bc] | 4.93 $\pm$ 0.34 [a] |
| N2 | 1.00 $\pm$ 0.00 [a] | 1.33 $\pm$ 0.05 [a] | 1.57 $\pm$ 0.07 [a] | 1.79 $\pm$ 0.04 [b] | 1.79 $\pm$ 0.08 [bc] | 2.38 $\pm$ 0.07 [b] | 3.27 $\pm$ 0.22 [ac] | 4.47 $\pm$ 0.52 [a] | 4.9 $\pm$ 0.21 [a] |
| D1 | 1.00 $\pm$ 0.00 [a] | 1.28 $\pm$ 0.06 [a] | 1.52 $\pm$ 0.05 [a] | 1.62 $\pm$ 0.03 [bc] | 1.90 $\pm$ 0.04 [b] | 2.06 $\pm$ 0.22 [bc] | 3.47 $\pm$ 0.43 [ac] | 3.89 $\pm$ 0.48 [ab] | 5.42 $\pm$ 0.55 [a] |
| D2 | 1.00 $\pm$ 0.00 [a] | 1.17 $\pm$ 0.04 [a] | 1.39 $\pm$ 0.05 [a] | 1.80 $\pm$ 0.05 [b] | 1.91 $\pm$ 0.14 [b] | 1.93 $\pm$ 0.09 [bc] | 2.19 $\pm$ 0.13 [bc] | 2.70 $\pm$ 0.17 [c] | 3.97 $\pm$ 0.53 [a] |
| DN0.5 | 1.00 $\pm$ 0.00 [a] | 1.26 $\pm$ 0.05 [a] | 1.32 $\pm$ 0.04 [a] | 1.36 $\pm$ 0.10 [c] | 1.41 $\pm$ 0.01 [c] | 1.41 $\pm$ 0.21 [c] | 2.47 $\pm$ 0.16 [bc] | 3.93 $\pm$ 0.30 [ab] | 4.45 $\pm$ 0.34 [a] |
| DN1 | 1.00 $\pm$ 0.00 [a] | 1.22 $\pm$ 0.05 [a] | 1.52 $\pm$ 0.10 [a] | 1.74 $\pm$ 0.18 [bc] | 2.00 $\pm$ 0.12 [b] | 2.24 $\pm$ 0.12 [bc] | 4.24 $\pm$ 0.25 [a] | 4.85 $\pm$ 0.30 [a] | 5.20 $\pm$ 0.62 [a] |

Weights of four replicates were recorded for each treatment. Pairwise comparison is conducted according to the Holm–Sidak method between different concentrations at the same cultivation time at $p \leq 0.050$ where the letters a–c explain the presence of significant differences. N1: 1 mg L$^{-1}$ NAA; N2: 2 mg L$^{-1}$ NAA; D1: 1 mg L$^{-1}$ 2,4-D; D2: 2 mg L$^{-1}$ 2,4-D; DN0.5: 0.5 + 0.5 mg L$^{-1}$ 2,4-D + NAA; DN1: 1 + 1 mg L$^{-1}$ 2,4-D + NAA.

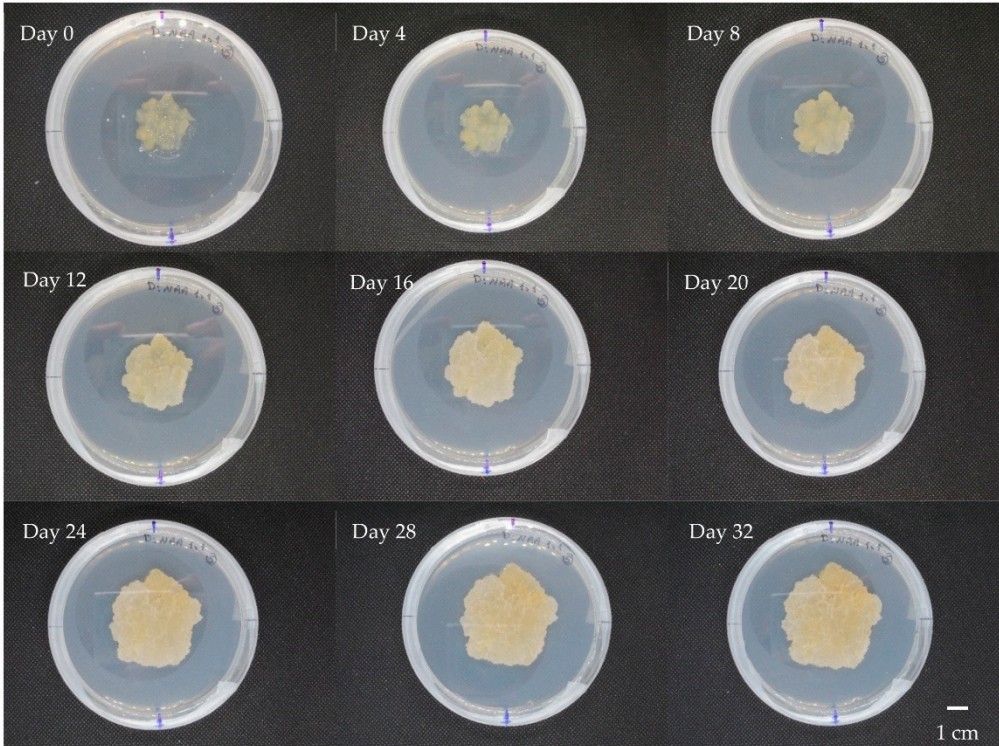

**Figure 3.** Dynamics of *L. schweinfurthii* callus growth during 32 days at 4-days intervals cultured on MS medium supplemented with 2,4-D and NAA of 1 mg L$^{-1}$ concentration each (DN1).

The changes in the size of the callus have been observed in terms of the callus area. The callus area in $cm^2$ was determined and calculated using ImageJ Software (ver. 1.53f51). The obtained results showed no significant differences in callus size during the first 8 days. The differences in callus size in the media containing a combination of 2,4-D and NAA (DN0.5 and DN1) were significant when compared with the other media types, starting from day 12 until the end of cultivation time. The callus multiplicated in DN1 medium showed the highest increase in callus area of 8.52 $cm^2$ within 24 days, while of 6.26 $cm^2$ in DN0.5 medium (Figures 3 and 4). The maximum increase in callus size was achieved in DN1 and DN0.5 after 32 days of 10.00 and 9.16 $cm^2$, respectively. The callus on NAA and 2,4-D media showed lower callus size, ranging between 5.71 $cm^2$ on N2 medium and 6.38 $cm^2$ on N1 medium. These results are in line with the results of fresh weight in terms of the best medium of callus multiplication. DN1 was the recommended medium for callus multiplication of *L. schweinfurthii* as it helped in producing the larger callus with the highest fresh weight and continually active dividable cells. Contrary to our current study, the mean callus diameter was used as a measure of callus enlargement of *Lycium barbarum* L. [67] and *Stevia rebaudiana* [68] instead of the actual size, where the callus diameter was determined by multiplying square root of callus length by callus width.

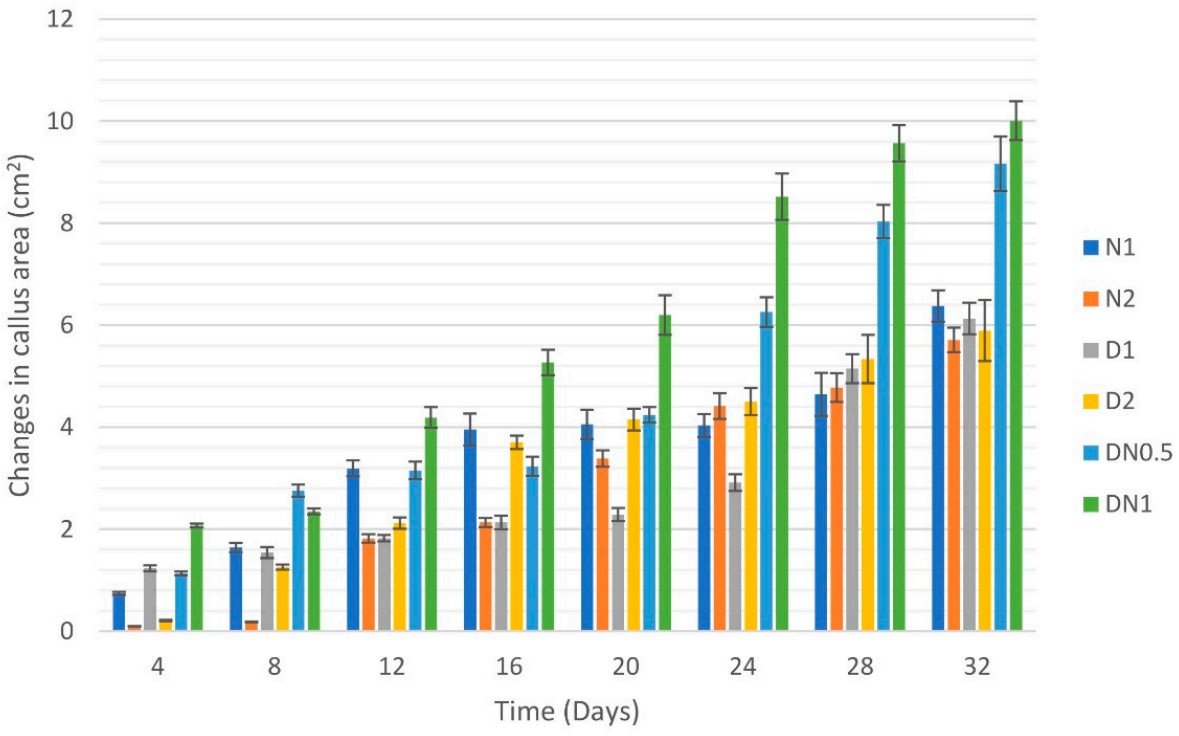

**Figure 4.** Changes in *L. schweinfurtii* callus size during 32 days of culture on MS medium supplemented with different concentrations of NAA and 2,4-D expressed in terms of callus area in $cm^2$. N1: 1 mg $L^{-1}$ NAA; N2: 2 mg $L^{-1}$; D1: 1 mg $L^{-1}$ 2,4-D; D2: 2 mg $L^{-1}$ 2,4-D; DN0.5: 0.5 + 0.5 mg $L^{-1}$ 2,4-D + NAA; DN1: 1 + 1 mg $L^{-1}$ 2,4-D + NAA.

*3.2. Cell Suspension Cultures*

Although the combination of 2,4-D and NAA was the best culture medium for callus multiplication, the use of liquid MS medium supplemented with 2 mg $L^{-1}$ NAA produces more reproducible cell suspension cultures than the 2,4-D and NAA combination. Therefore, different sucrose concentrations of 5, 10, 15, 20, 25, and 30 g $L^{-1}$ were applied to the chosen MS medium in which pH was adjusted at 5.7 to observe the differences in the fresh weight of the produced cells, the cell viability, and the changes in medium pH. After inoculation of 1 g friable callus in each 100 mL flask of each concentration, the cell cultures were incubated in a temperature-controlled shaker at 28 °C with a shaking rate of 120 rpm for 2 weeks. The four

replicates of each treatment were then vacuum filtered, weighed, and tested for viability, and the pH of the remaining media was measured. The fresh weights of cell cultures collected ranged between 3.80 and 12.01 g. The biomass produced in 25 and 30 g L$^{-1}$ sucrose media was the highest at 11.59 and 12.01 g, respectively. The lowest biomass of 3.80 g was obtained on a medium containing 5 g L$^{-1}$ sucrose. On the other hand, the medium pH was stable in the media containing 15 and 30 g L$^{-1}$ sucrose at 5.77 and 5.72, respectively. The medium pH increased above 6.00 in 5 and 10 g L$^{-1}$ sucrose media, while it decreased to 5.50 and 5.35 in 25 and 20 g L$^{-1}$ sucrose media (Figures 5 and 6). The sucrose concentration of 30 g L$^{-1}$ is commonly used in cell suspension cultures such as in cultures of *Artemisia absinthium* L. [69], *Ageratina pichinchensis* Kunth [70], and *Ophiorrhiza mungos* Linn. [71]. Otherwise, 50 g L$^{-1}$ sucrose was optimally used for enhanced production of secondary metabolites in *Withania somnifera* L. [72]. Furthermore, the type of PGRs and medium has a direct correlation with biomass and secondary metabolites production in cell suspension cultures. The outcomes vary from one plant species to another and should be studied separately [73–76]. The development of plant cell suspension cultures is a viable technology to produce plant metabolites. For example, the production of Taxol and ginsenoside through cell cultures is a model of commercial success in this area [77]. Moreover, the production of valuable secondary metabolites such as phenolics and flavonoids in cell suspension cultures opens the way for some plants to be introduced into the world market [78].

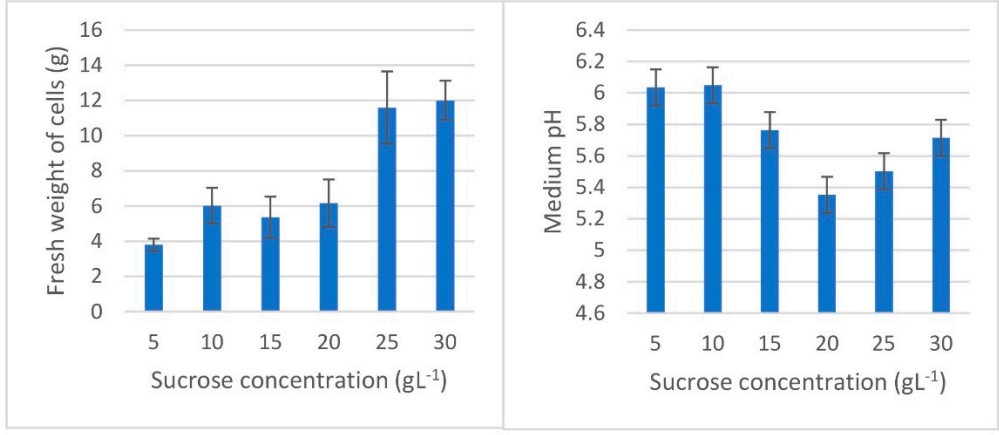

**Figure 5.** Fresh weights and medium pH of *L. schweinfurthii* cell cultures after 2 weeks of culture on 50 mL MS medium fortified with 2 mg L$^{-1}$ NAA. Inoculation weight was 1 g fresh callus per 50 mL liquid medium. Four replicates were used for each sucrose concentration.

The cell viability test was performed using 2,3,5-triphenyl tetrazolium chloride (TTC) for the produced cell cultures after 2 weeks to observe the cell cultures of higher viability. Six cell samples were taken from each sucrose concentration medium to test viability. The medium containing 5 g L$^{-1}$ sucrose was the only medium which had low viability, with an absorbance of ≤0.15. The normal viable cell cultures were found in all media of 10 g L$^{-1}$ sucrose or more (Figure S3). It was noticed that the cell viability increased as the sucrose concentration increased gradually in the studied concentrations, and the highest absorbance was obtained in the culture medium containing 30 g L$^{-1}$ sucrose of 0.488 (Table 3). To maintain the viability of cell cultures and their multiplication activity, it is necessary to subculture regularly every 2 weeks [79]. The sucrose concentration in the culture media maintains its optimal osmotic potential and thus the viability of cells. The supplement of standard sucrose concentration (30 g L$^{-1}$) in full strength MS medium is the best combination for biomass and secondary metabolites production in cell suspension cultures [80].

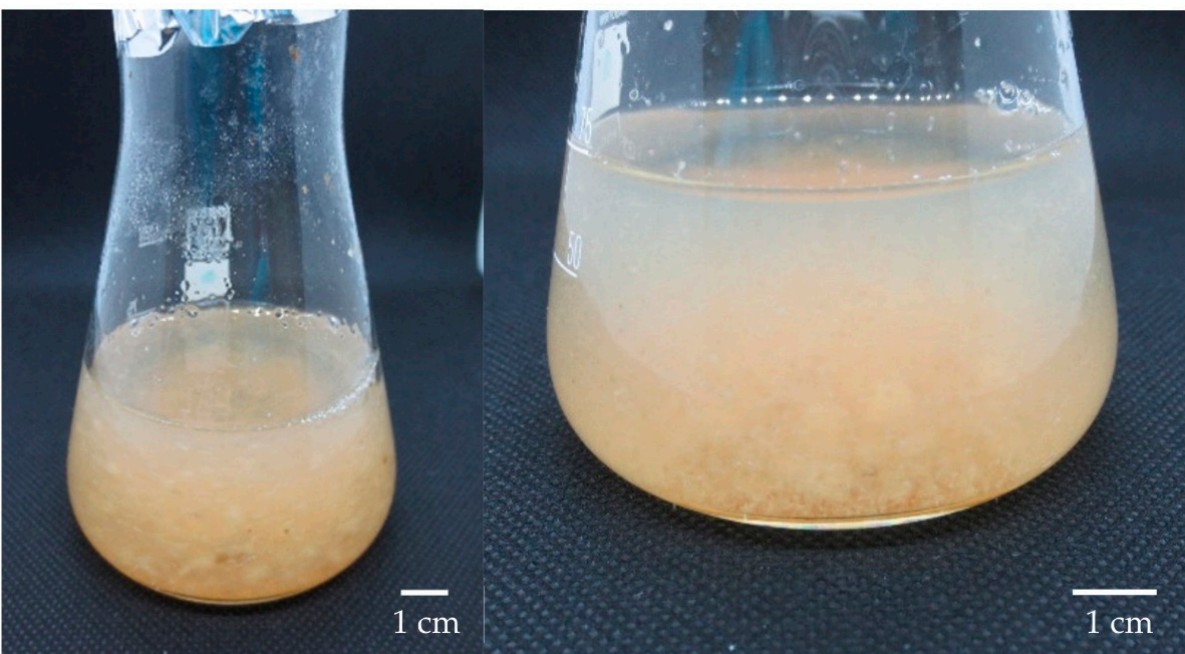

**Figure 6.** Cell suspension culture of *L. schweinfurthii* on N2 medium with 30 g L$^{-1}$ sucrose after 9 days of culture.

**Table 3.** Cell viability test of *L. schweinfurthii* cell suspension cultures on MS media with different sucrose concentrations after 2 weeks of culture.

| Sucrose Concentration (g L$^{-1}$) | Absorbance at 500 nm (Mean ± SE) | Viability Degree |
|:---:|:---:|:---:|
| 5 | 0.136 ± 0.010 [b] | Low viability |
| 10 | 0.308 ± 0.029 [ab] | Viable |
| 15 | 0.340 ± 0.030 [a] | Viable |
| 20 | 0.398 ± 0.086 [a] | Viable |
| 25 | 0.330 ± 0.027 [a] | Viable |
| 30 | 0.488 ± 0.043 [a] | Highest viability |

The letters a and b are showing the significant difference in the pairwise comparison according to Holm-Sidak method at $p \leq 0.050$.

### 3.3. Estimation of Phenolics, Flavonoids, and Antioxidant Capacity

The phenolic contents of the callus and cell cultures' extracts were estimated spectrophotometrically at 750 nm in terms of gallic acid. The generated equation for the gallic acid standard curve (Figure S4) was $y = 0.0042x + 0.0043$ $\left(R^2 = 0.9988\right)$. Regarding the effect of PGRs on *L. schweinfurthii* callus, the total phenolic contents, which were estimated in different callus multiplication media, showed no considerable differences. The amount of total phenolics ranged between 39.63 and 78.97 µg g$^{-1}$ fresh callus. The lowest content was on DN0.5 medium, while the highest was on N2 medium (Table 4). Otherwise, the different sucrose concentration media of *L. schweinfurthii* cell suspension cultures showed significant differences in the total phenolic content. The total phenolic compounds were highly significant on N2 media containing 30 and 5 g L$^{-1}$ sucrose over the other sucrose concentration media of 84.27 and 78.59 µg g$^{-1}$ fresh cells. There were not any considerable differences found between the media containing 10–25 g L$^{-1}$ sucrose. The lowest phenolic content estimated in cell suspension cultures was found in the N2 medium containing 15 g L$^{-1}$ sucrose of 22.19 µg g$^{-1}$ fresh cells (Table 5).

**Table 4.** Total phenolic, flavonoid contents, and antioxidant capacity in callus cultures of *L. schweinfurthii* showing the effect of different PGRs.

| Media | Total Phenolics (µg g$^{-1}$) | Total Flavonoids (µg g$^{-1}$) | DPPH (%) | ABTS$^+$ (%) |
|---|---|---|---|---|
| N1 | 64.87 ± 3.61 [a] | 12.90 ± 1.20 [b] | 52.44 ± 2.09 [ab] | 50.79±3.00 [a] |
| N2 | 78.97 ± 10.12 [a] | 24.43 ± 4.24 [a] | 68.19 ± 10.65 [a] | 49.74±3.81 [a] |
| D1 | 65.37 ± 12.48 [a] | 10.10 ± 0.42 [b] | 17.05 ± 4.03 [b] | 56.84±5.95 [a] |
| D2 | 71.10 ± 17.64 [a] | 10.49 ± 0.99 [b] | 22.21 ± 5.19 [ab] | 61.65±7.28 [a] |
| DN0.5 | 39.63 ± 3.44 [a] | 9.10 ± 0.36 [b] | 7.18 ± 1.45 [b] | 39.73±2.57 [a] |
| DN1 | 63.56 ± 6.90 [a] | 10.61 ± 1.02 [b] | 39.12 ± 10.18 [ab] | 58.73±6.22 [a] |

Concentrations of extracts were 500 mg mL$^{-1}$ for total phenolics and flavonoids estimation and 25 mg mL$^{-1}$ for antioxidant capacity determination. The letters a and b are showing the significant difference in the pairwise comparison according to Holm-Sidak method at $p \leq 0.050$.

**Table 5.** Total phenolic, flavonoid contents, and antioxidant capacity in cell suspension cultures of *L. schweinfurthii* showing the effect of different sucrose concentrations.

| Sucrose Concentration (g L$^{-1}$) | Total Phenolics (µg g$^{-1}$) | Total Flavonoids (µg g$^{-1}$) | DPPH (%) | ABTS$^+$ (%) |
|---|---|---|---|---|
| 5 | 78.59 ± 5.55 [a] | 13.71 ± 1.20 [b] | 33.19 ± 3.21 [b] | 27.43 ± 1.96 [b] |
| 10 | 44.64 ± 10.86 [b] | 9.56 ± 1.13 [bc] | 24.89 ± 3.17 [bc] | 23.32 ± 3.26 [b] |
| 15 | 22.19 ± 5.47 [b] | 9.47 ± 1.45 [bc] | 16.80 ± 0.67 [c] | 18.64 ± 1.93 [b] |
| 20 | 32.92 ± 2.83 [b] | 9.82 ± 0.52 [bc] | 32.53 ± 0.77 [b] | 21.65 ± 2.73 [b] |
| 25 | 33.70 ± 3.58 [b] | 8.36 ± 0.53 [c] | 32.40 ± 2.09 [b] | 29.85 ± 1.77 [b] |
| 30 | 84.27 ± 5.46 [a] | 19.14 ± 1.42 [a] | 49.46 ± 3.58 [a] | 56.29 ± 6.03 [a] |

Concentrations of extracts were 500 mg mL$^{-1}$ for total phenolics and flavonoids estimation and 25 mg mL$^{-1}$ for antioxidant capacity determination. The letters a–c are showing the significant difference in the pairwise comparison according to Holm-Sidak method at $p \leq 0.050$.

The total flavonoid contents were estimated spectrophotometrically in callus and cell cultures' extracts in terms of catechin at 510 nm. According to the standard curve of catechin, the equation of $y = 0.0034x - 0.0071$ $(R^2 = 0.9975)$ was generated (Figure S5). The callus cultures produced the highest content of flavonoids on MS medium supplemented with 2 mg L$^{-1}$ (N2) of 24.43 µg g$^{-1}$ fresh callus. The other PGRs media did not show any valuable differences with flavonoid contents ranging between 9.10 µg g$^{-1}$ in DN0.5 and 12.90 µg g$^{-1}$ in N1. On the other hand, the highly evaluated flavonoids in cell suspension cultures of *L. schweinfurthii* were on MS medium fortified with 2 mg L$^{-1}$ NAA (N2) and 30 g L$^{-1}$ sucrose of 19.14 µg g$^{-1}$ fresh cells followed by 5 g L$^{-1}$ sucrose of 13.71 µg g$^{-1}$ fresh cells. Lower flavonoid contents were found in media of 10–25 g L$^{-1}$ sucrose, ranging from 8.36 µg g$^{-1}$ in 25 g L$^{-1}$ sucrose media to 9.82 µg g$^{-1}$ in 20 g L$^{-1}$ sucrose media (Tables 4 and 5).

The obtained results revealed that the use of NAA or 2,4-D solely as in N1, N2, D1, and D2 stimulated almost similar amounts of phenolics when used in the media of callus multiplication, but the combination between them produced similar content of phenolics on higher concentration. Moreover, the flavonoid contents in both concentrations of media supplemented with NAA (N1 and N2) were elevated over other PGRs. Therefore, the supplement of NAA on callus multiplication media of *L. schweinfurthii* is preferred to produce flavonoids. Otherwise, it was noticed that low and high sucrose concentrations stimulated more phenolics and flavonoids production. This indicates the fact of sucrose importance in the osmotic pressure regulation in the culture media of *L. schweinfurthii* cell suspension cultures. The N2 medium of callus cultures and 30 g L$^{-1}$ sucrose medium of cell suspension cultures have the same constituents, except the N2 medium was solidified with Agar. It was found that this medium produced the highest phenolic and flavonoid contents in callus multiplication of 78.97 and 24.43 µg g$^{-1}$ and in cell suspension culture media of 84.27 and 19.14 µg g$^{-1}$, respectively. These results supported the use of 2 mg L$^{-1}$ NAA and 30 g L$^{-1}$ sucrose in the culture media of *L. schweinfurthii* callus and cell cultures

to produce higher contents of phenolics and flavonoids than that produced using other PGRs and sucrose concentrations used in this study. For the industrial production of plant natural sources, the plant cell and organ cultures provide a sustainable, controllable, and eco-friendly tool [81]. Several studies about callus and plant cell suspension cultures have been published in the last decade trying to introduce alternative sources of phenolic and flavonoid compounds. Robles-Martínez et al. [82] established callus and cell cultures of three *Opuntia* species to compare their potential as a source of in vitro metabolites. Among the species, in vitro cultures of *O. streptacantha* have a higher content of phenolic acids and antioxidant activity. In another study, it was found that the callus of *Zingiber officinale* Rosc. contained a lower amount of phenolics than that were found in the plant rhizomes but the exposure of callus to some elicitors enhanced the total phenolic compounds production significantly [83]. Furthermore, callus and cell suspension cultures of *Gymnosporia buxifolia* [84], *Thevetia peruviana* [85], and *Clinacanthus nutans* [86] were established to produce phenolic compounds in vitro where the phenolic compounds were successfully accumulated in the cultures.

The antioxidant activity of callus cultures' extracts was estimated by DPPH and ABTS$^+$ free radicals' methods to show the effect of PGRs type and concentration. The DPPH is known to be more sensitive and less specific than ABTS. The advantages of the DPPH assay are that it could be rapidly performed, it showed high reproducibility, and did not need time to generate free radicals as in the ABTS assay [87]. The highest antioxidant activity of callus cultures against DPPH free radicals has resulted in N2 medium of 68.19%, while the lowest was in DN0.5 medium of 7.18%. On the other hand, no significant differences were detected between callus against ABTS$^+$ radicals where the highest activity was D2, DN1, and D1 with 61.65, 58.73, and 56.84%, respectively. As in the DPPH results, the lowest antioxidant activity was on DN0.5 medium at 39.73% (Table 4). Comparing different sucrose concentrations in MS medium containing 2 mg L$^{-1}$ NAA, the highest significant differences in antioxidant capacity against DPPH and ABTS$^+$ free radicals were recorded in the 30 g L$^{-1}$ medium at 49.46% and 56.29%, respectively. Furthermore, the lowest antioxidant activity in cell suspension cultures was found on 15 g L$^{-1}$ of 16.80% and 18.64% against DPPH and ABTS radicals, respectively (Table 5). Although the DPPH method is supposed to be more sensitive in detecting slight differences in antioxidant activity, slow reactions between antioxidants and DPPH free radicals could result in some cases. This is maybe due to the fact that the solubility of DPPH is only in polar matrices [88]. Slight differences were detected in the antioxidant capacity between callus and cell cultures on the same medium with the same constituents (N2 medium in callus cultures and 30 g L$^{-1}$ sucrose cell cultures). Although the ABTS assay is superior to the DPPH assay for a variety of plant foods [89], our findings in this study support the use of both assays with the in vitro cultures of *L. schweinfurthii* because there was not dominant superiority of DPPH or ABTS assays found in these cultures.

## 4. Conclusions

In this study, we successfully established a suitable and efficient protocol for producing callus and cell suspension cultures of *L. schweinfurthii* to produce phenolics and flavonoids. The cultures' media were optimized in terms of the more suitable plant growth regulator and sucrose concentration for the higher biomass, phenolics, and flavonoids production. The MS medium fortified with 1 + 1 mg L$^{-1}$ 2,4-D + NAA was the best for increased callus cultures biomass and size, while that of 2 mg L$^{-1}$ NAA was the best for secondary metabolites accumulation. Due to their irregular shapes, the measurement of the actual size of the callus gives more precise results than the measurement of callus diameter. On the other hand, 2 mg L$^{-1}$ NAA media with 30 g L$^{-1}$ sucrose was the best for cell suspension cultures for higher biomass, phenolics, and flavonoids accumulation. Furthermore, the extracts of callus and cell suspension cultures could be used as alternative sources of antioxidants as the cultures produced promising quantities of polyphenols, particularly flavonoids. The antioxidative capacity of the extracts encourages introducing these in vitro cultures

to application. The use of more than one antioxidant assay in the in vitro cultures of the studied plant is recommended, especially DPPH and ABTS assays to obtain comprehensive results. The obtained results opened a path for the use of in vitro cultures of *L. schweinfurthii* for large-scale production of secondary metabolites in future studies.

**Supplementary Materials:** The following supporting information can be downloaded at: https://www.mdpi.com/article/10.3390/horticulturae8050394/s1, Figure S1: Leaf enlargement and new shoot formation of *L. schweinfurthii* on MS medium supplemented with 1 mg L$^{-1}$ BA after 7 weeks of culture using leaf explants; Figure S2: Side view of callus cultures of *L. schweinfurthii* multiplicated on MS medium supplemented with 2 mg L$^{-1}$ NAA; Figure S3: Microscopic examination of *L. schweinfurthii* cell cultures after incubation for 6 h with 1% TTC (2,3,5-triphenyl tetrazolium chloride) showing the red-violet stained living cells using light microscope; Figure S4: Gallic acid standard curve generated of different concentrations between 10 and 300 μg mL$^{-1}$ showing the absorbance measurements of 4 replicates at 750 nm measured by Folin-Ciocalteu assay; Figure S5: Catechin standard curve generated of different concentrations between 20 and 120 μg mL$^{-1}$ showing the absorbance measurements of 4 replicates at 510 nm measured by aluminum chloride assay.

**Author Contributions:** Conceptualization, methodology, software, validation, investigation, visualization, funding acquisition, D.M. and I.S.; statistical analysis, data curation, writing—original draft preparation, D.M.; resources, writing—review and editing, supervision, project administration, I.S. All authors have read and agreed to the published version of the manuscript.

**Funding:** D.M. was funded by a full scholarship (No. 308923) from the Ministry of Higher Education of the Arab Republic of Egypt. This Article is funded by the Open Access Publication Fund of Weihenstephan-Triesdorf University of Applied Sciences.

**Institutional Review Board Statement:** Not applicable.

**Informed Consent Statement:** Not applicable.

**Data Availability Statement:** Not applicable.

**Conflicts of Interest:** The authors declare that there is no conflict of interest.

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
