# Peer review of "Optimization of Callus and Cell Suspension Cultures of Lycium schweinfurthii for Improved Production of Phenolics, Flavonoids, and Antioxidant Activity"

_horticulturae, doi:10.3390/horticulturae8050394_

Round 1

Reviewer 1 Report

In general, the manuscript needs to be proofread to increase clarity.

Abstract OK

Introduction OK

Materials/Methods

2.1 The plant material that was used is seed and this should be avoided since seed is not true to type.

2.2 1 mm2? too small

2.5 why fresh sample? the reason for using fresh samples need to be explained since most of the studies used dried sample. 

Results

Basically most of the results is shallow and need more in depth data

2.9 and 2.10 Using percentage to express the value of antioxidants is not so accurate. Please use standard compounds such as Trolox so that same comparison can be done with other herb/plant/crop

Line 278, 279 avoid usage of simbol &

Line 335 2mg/L of what?

Discussion

The discussion was very brief and not well discuss/not in depth

Conclusion

No result was mentioned in the conclusion. Hence no conclusion on which treatment is good for callus/cell and production of phenolics/antioxidant can be found.

References

Some citations in the text does not follow the guidelines by MDPI. For eg line 92, 94, 169, 182, 193, 195, 300, 420

Author Response

Dear Reviewer,

Thank you very much for your valuable comments that should improve our manuscript. Please find the attachment.

Kind regards,

Diaa Mamdouh

Reviewer 2 Report

    Lycium schweinfurthii is a traditional medicinal plant grown in the Mediterranean region. As it is used in folk medicine to treat stomach ulcers, it took more attention as a source of valuable secondary metabolites. The in vitro cultures of L. schweinfurthii could be a great tool to produce secondary metabolites with low costs. The presented study aimed to introduce and optimize a protocol for inducing callus and cell suspension cultures as well as estimating phenolic, flavonoid compounds, and antioxidant activity in the cultures of the studied species. Three plant growth regulators (PGRs) were supplemented to MS medium solely or in combination to induce callus from leaf explants.The extracts of callus and cell suspension cultures could be used as alternative sources of antioxidants. The use of more than one antioxidant assay in the in vitro cultures of the studied plant is recommended, especially DPPH and ABTS assays to obtain comprehensive results.There are also some errors in the manuscript, as follows:

Major problem:

  1. The manuscript mentioned the extraction of phenols and flavonoids from Rhodiola rosea,Rosmarinus officinalis and other callus, why Lycium schweinfurthiiwas selected as research material, and what is the significance to actual production practice.
  2. The material method includes the determination of cell viability in suspension culture, and corresponding results charts should be included in the results and discussion.
  3. Estimates of phenolic compounds, flavonoids and antioxidant capacity should be graphically detailed, such as catechin measurements.
  4. There are many repetitions of the number and biology of various media in the manuscript, so it is suggested to add tables for expression.

Minor problem:

  1. Line 312Figure 3, The definition of callus images needs to be improved, so original images are recommended.
  2. Some bar charts in the manuscript do not have horizontal and vertical coordinates, so the font and size need to be unified.
  3. The scale of the pictures in the manuscript can be deleted, plus the size containers used in the experiment.
  4. Table 1, Table 2, Table 3 and Table 4 in the manuscript should be in the same format to ensure the same length.

Author Response

Dear Reviewer,

Thank you for your comments. Please see the attachment.

Best regards,

Diaa Mamdouh

Reviewer 3 Report

the conclusion part is kept very short, this part is the most important part of the article, it can be kept a little wider, the places that are considered important from the numerical data should be emphasized here so that the judgment sentences have a value

-in vitro checked,  in vitro

Author Response

Dear Reviewer,

Thank you very much for your valuable comments that will improve our article. Your comments were covered. The comments in detail are as follows:

*the conclusion part is kept very short, this part is the most important part of the article, it can be kept a little wider, the places that are considered important from the numerical data should be emphasized here so that the judgment sentences have a value.

We improved the conclusion as you recommended as follows:

In this study, we successfully established a suitable and efficient protocol for producing callus and cell suspension cultures of L. schweinfurthii to produce phenolics and flavonoids. The cultures’ media were optimized in terms of the more suitable plant growth regulator and sucrose concentration for the higher biomass, phenolics, and flavonoids production. Furthermore, the extracts of callus and cell suspension cultures could be used as alternative sources of antioxidants as the cultures produced promising quantities of polyphenols, particularly flavonoids. The antioxidative capacity of the extracts encourages introducing these in vitro cultures to application. The use of more than one antioxidant assay in the in vitro cultures of the studied plant is recommended, especially DPPH and ABTS assays to obtain comprehensive results. The obtained results opened a path for the use of in vitro cultures of L. schweinfurthii for large-scale production of secondary metabolites in future studies.

*-in vitro checked,  in vitro

in vitro checked and italicized in the whole manuscript.

Sincerely,

Diaa Mamdouh

Reviewer 4 Report

I am interested in the results of the study entitled ‘Optimization of callus and cell suspension cultures of Lycium Schweinfurthii for improved production of phenolics, flavonoids, and antioxidant activity’. I inform you that it can be printed for the following reasons.

The tissue culture of Lycium Schweinfurthii well selected and the callus successfully induced. By creating an appropriate ratio of NAA and 2,4-D, we succeeded in stimulating callus production. It is a great development to produce antioxidant-related substances through liquid suspension culture of callus in N2 medium.

Author Response

Dear Reviewer,

We would like to thank you a lot for your kind comment that moves us forward.

Sincerely,

Authors

Reviewer 5 Report

Dear Author,

The manuscript entitled “Optimization of Callus and Cell Suspension Cultures of Lycium Schweinfurthii for Improved Production of Phenolics, Flavonoids, and Antioxidant Activity” was revised. It was a very well prepared and very laborious work. In general, the discussion can be expanded a little more. The present study should be supported by referencing similar studies of literature. Therefore, the "Results and discussion" section should be improved. The manuscript can be published after several corrections and some improvements.

This corrections and suggestions described below;

Title

OK

 Abstract

Line 12, “in vitro” should be written in italics.

 Keywords

Line 30-31, keywords should be given in alphabetical order. At the same time, it is more appropriate to give other words as keywords instead of the words in the title of the article.

 Introduction

Line 34, Genus and species names should be written in italics…. “Lycium

Line 35, ….“Lycium”…

Line 39, … “L. schweinfurthii”…. Since the same species name is mentioned, it is not necessary to write the full genus name…. “L.” is enough

Line 54-55, If this sentences “Successful plant tissue culture depends on many factors, particularly plant growth regulators (PGRs), the age of tissue, the type of medium, and the type of used explant [11, 12]” comes after “ ………….exploring alternative pharmaceuticals from plants [13].” , it will more meaningfull.

Line 83, .. “in vitro” … italics

Line 99,.. ….“L. schweinfurthii”….

Material Methods

Line 105, … “Clorox”  is this “bleach” or a bleach brand ..?

Line 110, …. It would be better if it was given as light intensity instead of “2500 lux”, for example, 50 μmol−1 m−2 s-1

Line 117-119, “….2,4-D (0.5, 1, 2 mg L-1), NAA (0.5,1, 2 mg L-1), BA (0.5, 1, 2 mg L-1), BA+2,4-D (1+0.5, 1+1, 1+2 mg L-1, respectively), BA+NAA (1+0.5, 1+1, 1+2 mg L-1, respectively) or 2,4-D+NAA (0.5+0.5, 1+1, 2+2 mg L-1)…. It seems like it would be better if it was given as a table to make it more understandable.

Results and Discussion

Line 232, … “(Murashige and Skoog)” … "Materials and methods", where it was first mentioned, can also be cited as a source.

Line 257, …and the mass propagation of …..

Line 278-279, ….why did you use “&”

Line 322, … “Figure 3 & 4” ??? ………..“3, 4”

Line 349, …. “Figure 5 & 6” ??? ………..“5, 6”

Line 397, …the value of “13,71”, ….The numeric values in the whole text are separated by "dot", here is "comma". … “13.71”

Line 428, “in vitro” … italics

Line 458, “in vitro” … italics

In general, the results are given in full in this section, but the discussion can be expanded a little more. Because the results obtained should be associated with some studies in the literature, and the difference and benefit of the study should be supported. The present study should be supported by referencing similar studies of literature. Therefore, the "Results and discussion" section should be improved. For example, in line 334-354, Subheading “Cell suspension cultures” is almost never discussed.

Conclusions

OK

 References

OK

Author Response

Dear Reviewer,

Thank you for your detailed comments. They were really helpful to find out the negative points in our manuscript. All of your comments were taken into account cautiously and deeply. Your comment about the discussion is greatly appreciated. Please see the attachment.

Kind regards,

Diaa Mamdouh

Round 2

Reviewer 1 Report

Accepted after correction